# Are Australian Universities Perpetuating the Teaching of Racism in Their Undergraduate Nurses in Discrete Aboriginal and Torres Strait Islander Courses? A Critical Race Document Analysis Protocol

**DOI:** 10.3390/ijerph19137703

**Published:** 2022-06-23

**Authors:** Keera Laccos-Barrett, Angela Elisabeth Brown, Roianne West, Katherine Lorraine Baldock

**Affiliations:** 1UniSA Clinical and Health Sciences, University of South Australia, Adelaide, SA 5000, Australia; angela.brown@unisa.edu.au; 2College of Nursing and Health Sciences, Flinders University, Adelaide, SA 5042, Australia; 3Rosemary Bryant AO Research Institute, UniSA Clinical and Health Sciences, Adelaide, SA 5000, Australia; 4Congress of Aboriginal and Torres Strait Islander Nurses and Midwives, Murarrie, QLD 4172, Australia; ceo@catsinam.org.au; 5Workforce Innovation, Charles Darwin University, Darwin, NT 0810, Australia; 6Teaching Innovation Unit, University of South Australia, Adelaide, SA 5000, Australia; katherine.baldock@unisa.edu.au

**Keywords:** Australian Aboriginal, health inequity, racism, race, nursing, education, curriculum

## Abstract

Systemic racism has a profound negative impact on the health outcomes of Australia’s First Nations peoples, hereafter referred to as Aboriginal and Torres Strait Islander peoples, where racism and white privilege have largely become normalised and socially facilitated. A national framework is being mobilised within the tertiary-level nursing curriculum to equip future health professionals with cultural capabilities to ensure culturally safe, equitable health care for Aboriginal and Torres Strait Islander peoples. In 2019, nurses comprised more than half of all registered health professionals in Australia, and current national standards for nursing state that Australian universities should be graduating registered nurses capable of delivering care that is received as culturally safe. It is therefore critical to evaluate where learning objectives within nursing curricula may lead to the reinforcement and teaching of racist ideologies to nursing students. This protocol outlines a framework and methodology that will inform a critical race document analysis to evaluate how learning objectives assert the social construction of “race” as a tool of oppressive segregation. The document analysis will include each discrete Aboriginal and Torres Strait Islander health course within all undergraduate nursing programs at Australian universities. The approach outlined within this protocol is developed according to an Indigenous research paradigm and Colonial Critical Race Theory as both the framework and methodology. The purpose of the framework is a means for improving health professional curriculum by reducing racism as highlighted in nation-wide strategies for curriculum reform.

## 1. Introduction

As of 2020, Australia’s First Nations peoples, hereafter referred to as Aboriginal and Torres Strait Islander peoples, face a burden of disease that is 2.3 times that of non-Indigenous Australians [1]. Aboriginal and Torres Strait Islander peoples are dying from avoidable causes at three times the rate of non-Indigenous Australians [2].Yet, while Aboriginal and Torres Strait Islander peoples are twice as likely to attend hospital, they receive lower levels of treatments and procedures compared to non-Indigenous Australians [2]. A review by Paradies [3] found that Aboriginal and Torres Strait Islander peoples were less likely than non-Indigenous Australians to receive early diagnosis and treatment for various conditions including cancers (lung; cervical cancer screening, diagnosis, and treatment; breast; head and neck; general cancer; cancer survival), coronary procedures, and kidney transplantation. These discrepancies between hospital attendance, lower levels of treatment, and poorer health outcomes raise concerns about racism deeply embedded within the Australian healthcare system.

Racism is experienced by Aboriginal and Torres Strait Islander peoples when accessing healthcare and occurs both interpersonally and institutionally [4]. Racism itself can be best defined as by Lorde [5]: “the belief in the inherent superiority of one race over all others and thereby the right to dominance”. Durey [4] stated that racism is a social determinant of health where racism is embedded in social, structural, and political contexts both at a personal level and also at an institutional level. A common definition of institutional racism, also known as systemic racism [3], is described by Macpherson [6] as:


*“The collective failure of an organization to provide an appropriate and professional service to people because of their color, culture, or ethnic origin. It can be seen or detected in processes, attitudes and behavior which amount to discrimination through unwitting prejudice, ignorance, thoughtlessness and racist stereotyping which disadvantage minority ethnic people.”*


Health professionals occupy extensive and powerful roles within the health system as “caregivers, communicators and educators, team members, managers, leaders, policy makers and knowledge brokers” [7]. Frenk et al. [7] discussed the need for health professional education reform in order to develop transformational service providers who adapt to adjustments in global health needs. Such sentiments echo the position of the World Health Organization (WHO) [8]: that reducing health inequalities requires interventions that empower populations through systemic change. This is imperative, as when accessing health services, Aboriginal and Torres Strait Islander peoples experience racism in several ways, such as interpersonally, through health minimization and misdiagnosis [2,9], and systemically, such as perceived cultural superiority by health professionals, which impacts health policy, healthcare conditions, and practices of institutions and staff [1,10,11,12]. Such experiences are contradictory to many articles of the United Nations Declaration on the Rights of Indigenous Peoples [13], highlighting a dichotomy between Aboriginal and Torres Strait Islander “rights” and reality while accessing services provided by health professionals.

A systematic review [14] found that a majority of research investigating racist beliefs of healthcare professionals has been performed in the United States (US). Two-thirds of the studies included in the review found evidence of racism, including racist beliefs, emotions, or practices by health professionals. Australian studies, however, have been limited to health professionals’ understandings of systemic racism, with findings demonstrating that it is poorly understood [15,16]. 

Wilson [17] found that non-Indigenous Australian health professionals had varying levels of awareness of the impact of colonization, with many not understanding the intergenerational legacy and impact on health and wellbeing. How such limited understanding of systemic racism translates into health accessibility is shown in an audit of 12 the South Australian Local Health Networks [18]. This audit scored evidence of institutional racism against five key indicators: inclusion in governance, policy implementation, service delivery, employment, and financial accountability. Scores higher than 160 were indicative of low levels of institutional racism, whereas scores <39 were reflective of very high levels of institutional racism. Of the ten health networks assessed, nine scored high levels of institutional racism with ratings of 39.5 or under [18].

Systemic racism in the Australian health care system is reflected in several ways: fragmented funding, which exercises tight control over Aboriginal Community Controlled Health Services [19]; lower levels of benefits comparable to non-Indigenous Australians arising from policies still impacting families, as outlined in *the Bringing Them Home* report [20]; and policy failures of the *Closing*
*the Gap in Aboriginal and Torres Strait Islander Health Inequity* as described by [21]. Given that health professionals occupy powerful roles, such as policy makers and leaders, it is imperative to question the role they play in enabling and perpetuating racism and how this can be transformed for the sake of health equity.

Nurses, specifically, occupy a considerable portion of the health professional workforce, in 2019 making up more than half of the total registered health professionals in Australia [22]. As key brokers in health care systems, the International Council of Nurses (ICN) Code of Ethics for Nurses [23] maintains that nurses share responsibility for initiating and supporting actions to meet both the social and health needs of the public. This includes challenging unethical organizational environments [23]. Such sentiments are shared nationally in professional codes and standards by the Nursing and Midwifery Board of Australia [24,25] and also strongly by the Australian Health Practitioner Regulation Agency (AHPRA) in the recent publication of the *National Scheme’s Aboriginal and Torres Strait Islander Health and Cultural Safety Strategy 2020–2025* [26]. 

The National Scheme’s Aboriginal and Torres Strait Islander Health and Cultural Safety Strategy 2020–2025 aims to make cultural safety the norm for Aboriginal and Torres Strait Islander peoples, with Strategy 2 targeting education and training standards and accreditation guidelines [26]. Strategy 2 of the initiative involves the National Boards and their committee adopting and endorsing the Aboriginal and Torres Strait Islander Health Curriculum Framework [27]. This supports the stance taken by Universities Australia [28], the peak body representing the 39 Universities within Australia and their released Indigenous curriculum guidelines.

With respect to the education of registered nurses, the *Australian Nursing and Midwifery Accreditation Council* Registered Nurse Accreditation Standards for higher education [29] states that Australian universities should be graduating registered nurses who are capable of delivering care that is received as culturally safe, the antithesis of personal and structural racism. The term “cultural safety” originated in Aotearoa, New Zealand and stems from the work of Ramsden [30] as an approach to nursing education, helping those involved to become aware of their social conditioning and how it impacts their practices. Ramsden [30] outlined that “cultural safety is an outcome of nursing education that enables safe service to be defined by those that receive the service and is achieved when the recipients of care deem the care to be meeting their cultural needs”.

The social construction of “race” is used to differentiate cultural groups into positions of superiority over another, the perpetuation of which can be included within higher education curriculum [31,32]. The Aboriginal and Torres Strait Islander Health Curriculum Framework [27] was developed to support higher education providers in the development of Aboriginal and Torres Strait Islander health professional curriculum. It is structured to develop the graduate capabilities of respect, communication, safety and quality, reflection, and advocacy [27]. The graduate capabilities are aligned to learning objectives, with examples provided in the framework [27]. Learning objectives (also known as learning outcomes) form the basis of the unit of study (course) and guide the curriculum design and development within programs (degrees) such as a Bachelor of Nursing; they describe what students are expected to understand and learn from the course and provide the basis for the development of student assessments [27]. The framework centres on eight principles underpinning the conceptual design of curriculum, model of implementation and context for successful [27]. The Registered Nurse Accreditation Standards by the Australian Nursing and Midwifery Accreditation Council [29] now require a “discrete Aboriginal and Torres Strait Islander health course” based on the Nursing and Midwifery Aboriginal and Torres Strait Islander Health Curriculum Framework [33]. This document is a Nursing and Midwifery adaptation of the Aboriginal and Torres Strait Islander Health Curriculum Framework [27]. The Nursing and Midwifery adaptation builds on the initial framework and is targeted at pre-registration courses for registered nurses and registered midwives [33]. As such, the learning objectives developed in the [33] framework are realigned to a nursing and midwifery context. Given what is known about racism in the Australian health care system and the need for culturally safe health practitioners, it is critical to evaluate where learning objectives within nursing curricula may be reinforcing and therefore guiding the (implicit or explicit) teaching of racist ideologies to nursing students as opposed to educating a workforce that provides care anti-racist care, which can be experienced as culturally safe.

A critical issue in evaluating learning objectives is the identity of the writer of those objectives and related curriculum. With the support of guidelines and frameworks, such as that of the Aboriginal and Torres Strait Islander Health Curriculum Framework [27], a modicum of consistency can potentially be achieved in national curriculum. This is particularly relevant when questioning the ability of graduates to meet the anti-racist sentiments outlined in standards and codes for the registered nurse [23,24,25]. However, there is subjectivity in who writes, develops, and populates content for Aboriginal and Torres Strait Islander health courses based on the positionality and understanding of the person who is developing the course [34,35]. Gillborn [36] emphasized that even well-intentioned actions can have racist outcomes through the minimization of the impact of colonization, othering, and attributing inequity to factors other than that of racism. In fact, deficit discourses in Indigenous education are still predominant in curriculum content [37,38,39,40,41]. In such ways, “race” can be employed as a tool that facilitates the beliefs that one group in society is superior to another, resulting in “race” oppression [42,43]. Where curriculum is designed to promote anti-racist ideals and develop future health professionals to enable culturally safe health services, it is imperative to critically analyse the objectives guiding the content. Specifically, how do learning objectives within discrete Aboriginal and Torres Strait Islander people’s health courses in undergraduate nursing programs at Australian Universities assert the social construction of “race” as a tool of oppressive segregation?

## 2. Materials and Methods

The study reviewed the learning objectives from all discrete Aboriginal and Torres Strait Islander health courses within Australian undergraduate nursing programs. Specifically, the authors determined how race is used a tool to oppressively segregate Aboriginal and Torres Strait Islander peoples in the course learning objectives and leads to the reinforcement and teaching of racist ideologies. The learning objectives were reviewed by applying document analysis and interpreted through an Indigenous research paradigm and Colonial Critical Race Theory (ColonialCrit) as both the methodology and theoretical framework.

### 2.1. Research Methodology

My The Indigenous research paradigm was developed based on the Indigenist research methodology by Lester-Irabinna Rigney [44]. This is to assert my Aboriginality as a Ngarrindjeri woman from the Coorong region of South Australia into my research to maintain accountability to my communities. The Indigenous research paradigm informs the way I view reality (ontology), the way that I think about reality (epistemology), and my guiding ethics and morals (axiology) together [45,46,47,48]. Many of the dominant paradigms do not align with the axiology of an Indigenous way of knowing; however, ColonialCrit does, and it has been developed within an Aboriginal and Torres Strait Islander context [42]. ColonialCrit works to four tenets that recognize the social embeddedness of racism, assert the social construction of race as a tool of oppressive segregation, privilege counter-story telling, and commit to social justice and praxis [42]. Where the privileging of counter-story telling is a primary tenet of ColonialCrit, I assert that the use of an Indigenous research paradigm (the ontology, epistemology, and axiology) is imperative if such a study is to truly explore raciology from an Aboriginal and Torres Strait Islander way of knowing and being. The research project is informed by an Indigenist Research methodology [44] and Colonial Critical Race Theory, which together act as both the methodology and the theoretical framework.

The Indigenous research paradigm, The Indigenist Research Methodology, developed by Australian Aboriginal scholar Lester Irabinna Rigney [44] is guided by the following principles:Resistance as the emancipatory imperative of Indigenous research: the Aboriginal and Torres Strait Islander struggle for resistance from racist oppression whilst seeking to uncover and protest continuing forms of oppression;Political integrity in Indigenous research: Aboriginal and Torres Strait Islander research, which is undertaken by Aboriginal and Torres Strait Islander peoples. This principle emphasizes that Aboriginal and Torres Strait Islander peoples set our own agenda for political liberation, and we are responsible to our communities and their struggle;Providing Indigenous voices in research: research that focuses on the lived, historical experiences, ideas, traditions, dreams, interests, aspirations, and struggles of us [44].

As Wilson [45] explained, Guba and Lincoln [49] outlined four dominant paradigms (positivism, post positivism, critical theory, and constructivism), all of which share the perspective that knowledge is individual and thereby subject to individual interpretation and bias. Contrary to these, within an Indigenous paradigm, knowledge is not owned by the individual; we are one and of the cosmos [45,46,48]. In this regard, I challenge the ability of a non-Indigenous research paradigm to understand “race” as a tool of oppressive segregation upon Aboriginal and Torres Strait Islander peoples if it is not explored through Aboriginal and Torres Strait Islander ways of knowing and being. As such, what would be considered my bias in a non-Indigenous paradigm is essential to work within an Indigenous research paradigm. Likewise, while non-Indigenous researchers can partake in Indigenous analytics, the Indigenous research per se cannot be reproduced due to the very nature of our Indigenous-embodied knowledges [46]. The rationale behind building my research from this Indigenous research paradigm is my existential relationship with the cosmos and my responsibility and accountability to the struggles of myself and my people. Resistance, integrity, and privileging our voices is at the forefront of this positionality, that is, the unmasking of racist oppressions that continue to be part of our reality [44].

### 2.2. Colonial Critical Race Theory 

Framing this research within an Indigenous research paradigm has determined my choice in using Colonial Critical Race Theory (ColonialCrit). This is largely due to the difficulty of separating non-Indigenous research tools from the underlying beliefs that created them [45]. Where many positivist and post-positivist paradigms fail to align with the axiology of an Indigenist research methodology, ColonialCrit does so in the political and resistive commitment to challenging racism and oppression [42,44,45].

Critical Race Theory (CRT) originated in civil rights litigation in the U.S. during the 1970s and was grounded in racialized experiences of people of colour, challenging the white discourse on “race” and its role in racial oppression [50,51,52,53]. CRT is governed by a set of tenets to guide the framework and beliefs on racism as shown in Gillborn [50]. CRT endeavours to reveal how racism is normalized and deeply embedded within society through hidden power dynamics and silences minoritized knowledges and experiences [50].

Since the 1990s, CRT has garnered much attention as a means to analyse dominant discourses of race in several fields including but not limited to Latino Critical Race Studies [54], Tribal Crit [55,56], and AsianCrit [57]. CRT has also been applied to disciplines as a tool for challenging race-based inequalities in education as initially developed by Ladson-Billings [58]. This was then further expanded into higher education [31,59] and later refined to address experiences of Indigenous peoples in education within the U.S., which foregrounds the discourse of colonization, imperialism, Indigenous identities, sovereignty, assimilation, tribal philosophies, and stories as theory and social change [55]. Such works have been built upon by researchers into the Australian educational context [42,60,61,62,63].

Australian Indigenous scholar Moodie [41] argued that where CRT has not been modified for an Indigenous context, it will not address placed-based histories of resistance and struggle, emancipation and success, or meanings and traditions of the specific communities [64]. In this study, failure to modify CRT to an Australian context will compromise my Indigenous research paradigm, as there is a need to centralize Indigenous voices in research integrity and political resistance. Nor will an unmodified version of CRT factor in that the context from which the theory was initially developed does not necessarily equate to racism as experienced within another context [65].

An Australian adaptation of CRT, Colonial Critical Race Theory (ColonialCrit), was developed by Kamilaroi woman Sheelagh Daniels-Mayes [42], which centres the stories and histories of Aboriginal and Torres Strait Islander peoples. Where the Australian educational CRT adaptation by Moodie [41], *Decolonising Race Theory*, is focused on decolonizing, ColonialCrit incorporates both raciologies and decolonizing theories as represented in the research question [42]. In addressing race-based inequalities in Australian universities to simply ‘be good’ and ‘not racist’ lends towards institutional passivity [66]. However, this can be difficult where Australia universities further marginalize, denigrate, and exploit Aboriginal and Torres Strait Islander voices through silencing [67]. The challenges of which are evident where partnerships with Aboriginal and Torres Strait Islander peoples are suggested, but there is still limited transparency and accountability to drive evaluation and improvements in health [68,69]. ColonialCrit amalgamates seven prominent CRT tenets into four [42]. The tenets are as follows: Recognizing the social embeddedness of racism;Asserting the social construction of race as the tool of oppressive segregation;Privileging of stories and counter-story telling;Committing to social justice and praxis: incorporating activism.

The first tenet establishes that racism is and has been deeply embedded in Australia since colonization [42]; that racism is “naturalized”, and institutional racism is largely unchallenged; and that institutions adopt a colour-blindness approach, which, according to the dominant perspectives, asserts that racism does not exist [42]. As a means of incorporating this tenet into the research framework, I will establish that racism is embedded, as shown through a review of the literature with examples of racism in health care and the resulting inequitable health outcomes and access.

The second tenet explores how “race” as a social construction is used as a means to enable and employ power by one group in society over another group [42]. This is achieved through constructions of deficit, be it cultural, biological, or moral, which places the blame on individuals rather than the policies and practices that initiate and continue to enforce race-based inequality [42].

The third tenet of ColonialCrit is the privileging of stories and counter-story telling, challenging the dominant narrative that is underpinned by racism [42]. This focuses on privileging experiences and stories of Aboriginal peoples and having such accepted as legitimate and valuable by the dominant narrative [42]. The objective of counter-story telling in this instance is to disrupt the majoritarian, normalized, and racist narrative through highlighting examples within the learning objectives where a counter story is present [42].

The fourth and final tenet is the commitment to social justice and practice: incorporating activism [42], the commitment to transformation through the identification and challenging of normalized race-based barriers to social justice [42].

The research question for this study has been drawn directly from the second tenet of ColonialCrit: “Asserting the social construction of race as the tool of oppressive segregation” [42]. As the key proponent for the research question, the data analysis explores how the social construction of “race” is used to uphold inequitable power dynamics

The objective of the research project in highlighting where and how race is a tool of oppression is to facilitate transformational change to curriculum and the resulting graduate health professionals. Ultimately, this will assist in identifying and challenging institutionally racist education, thereby contributing to the goal outlined in strategy 2 of The National Scheme’s Aboriginal and Torres Strait Islander Health and Cultural Safety Strategy 2020–2025 [26], to ensure culturally safe and accessible care for Aboriginal and Torres Strait Islander peoples.

## 3. Results

### 3.1. Analysis

Data analysis will occur through an interview method adopted from the questions developed by Gillborn [50] for education policy and Whiteness by Leonardo [70].

#### Data Analysis Framework

A document analysis is a systematic procedure for evaluating documents that requires text or images to be examined and interpreted to gain meaning, understanding, and knowledge [71]. A document analysis is differentiated from a literature review in that the text itself is the primary source of data informing a primary research project rather than secondary data informing a review of primary research [72]. In the research project, a document analysis of discrete Aboriginal and Torres Strait Islander peoples’ health undergraduate nursing course learning objectives will be performed. An eight-step planning process by O’Leary [72] for textual analysis will be followed in the research project (Figure 1):

Document analysis is iterative and combines the use of both content analysis and thematic analysis [71]. A key consideration for data analysis in this project is that “race” as a social construct is fluid. It is impractical to create a list of ways racism is used to oppress and segregate, as the methods to do so are ever evolving [50]. As Gillborn [50] stated, there is an absence of a clear conceptual map of anti-racism. With this in mind, a questionnaire proforma was produced based on the questions by Gillborn [36] and Leonardo [70]. 

The limitations of a document analysis are identified as low retrievability and biased selectivity [71]. If low retrievability becomes a research barrier, based on step 8 of the O’Leary guide [72], a contingency plan to overcome such a barrier would be to shift the research focus to the “course aims” rather than the learning objectives, as they are publicly accessible. This would, however, mean that the assessable learning objectives that guide the course structure would not be the focus of the study. Another limitation to document analysis is bias selectivity [71]; however, this is overcome, in this instance, as eligible nursing programs will be drawn from a list provided by the Australian Health Practitioner Regulation Agency [73] and cross-referenced by manual searching.

Additional limitations of document analysis are identified as author bias and researcher bias [72]. The O’Leary [72] framework step 3 calls for identification of bias, including whether the text was edited or anonymous. To position the data in terms of potential author bias, the first question in data analysis concerns authorship transparency: “Who wrote the learning objectives?”. The credibility of the data that I will generate will be dependent on the recognition of author bias [72]. This is also very significant regarding the socially constructed power dynamic, including who writes and controls the discourse on Aboriginal and Torres Strait Islander peoples. Regarding my own researcher bias, I have previously established why my positionality as a Ngarrindjeri woman is imperative to an Indigenous research paradigm and how this enables a counter-story-telling dialogue with the data as per tenet 3 of the ColonialCrit framework. Additionally, it is imperative to reflect on the biases presented by a White lens critique, which often passes over Indigenous knowledges, as a tool of race-based power.

### 3.2. Inclusion Criteria

Course learning objectives are restricted to discrete Aboriginal and Torres Strait Islander health courses within Bachelor of Nursing degrees that are recognised as approved programs of study by the Australian Health Practitioner Regulation Agency. Where there are dual degrees, for instance, Bachelor of Nursing/Bachelor of Midwifery, the programs will be scrutinised to identify if the discrete Aboriginal and Torres Strait Islander peoples’ health course is the same course that is delivered in the Bachelor of Nursing alone. Where this is the case, duplicates will be removed. Only the discrete Aboriginal and Torres Strait Islander courses that are mandatory for accreditation purposes by ANMAC [29] for Bachelor of Nursing programs will be included. Courses within nursing programs that are not approved programs of study will not be included, programs that are not delivered by Australian universities will similarly not be included. Post-graduate programs, bridging programs, and re-entry programs will not be included.

#### Sample

The sample will be collected from the Australian Health Practitioner Regulation Agency search tool, which provides a list of approved programs at the following address: https://www.ahpra.gov.au/Accreditation/Approved-Programs-of-Study.aspx (accessed on 3 March 2021). The discrete Aboriginal and Torres Strait Islander health courses will be identified by reviewing the relevant program of study information for each university on their websites. For those universities that do not make their learning objectives publicly available, the universities will be contacted for copies of their learning objectives in their Aboriginal and Torres Strait Islander discrete health courses for the nursing programs. Texts for data analysis will be collected from the discrete Aboriginal and Torres Strait Islander health course information page via the university website.

### 3.3. Data Analysis

Document analysis involves skimming, reading, and interpreting data through combining elements of both thematic and content analysis [71]. In line with tenet 3 of ColonialCrit [42], counter-story telling, the objective of the document analysis will be to challenge the racist, dominant narrative. This will be achieved through first highlighting data that provide a counter story guided by the data analysis tool (Appendix A) [71], followed by identifying patterns within the data and emerging themes (i.e., thematic analysis) that become categories of information for analysis [71].

A systematic approach for analysis will be adopted as documented by O’Leary [72]. This includes familiarising with the appropriate software for analysing qualitative data; entering data in a systematic way based on pre-defined protocols; organising data according to the source; reading through and taking overarching notes to guide the further iterations of sorting and categorising the data where the notes form part of the analysis; and finally, preparing analysis. 

Once the data are entered into NVivo, the document analysis will involve the identification of bias according to the interview questions; reducing, organising, and coding data; searching for patterns and interconnection; mapping and building themes; building and verifying theories; and drawing conclusions [72].

#### Data Analysis Tool 

Data analysis coding will be guided by an interview proforma informed by the works of Gillborn [50] and Leonardo [70]. Gillborn [50] developed three questions to assess educational policy in the U.S., which have been slightly modified for the purposes of this project. They are:Who wrote the learning objectives? This positions the authorship of the learning outcomes to identify potential bias as per the document analysis framework [72];Who wins and who loses based on the priorities of the objectives? Does the learning objective seek to empower Aboriginal and Torres Strait Islander peoples, or does it further empower nurses towards paternalism, prejudice, low expectations, and White saviorism? Empowerment of Aboriginal and Torres Strait Islander peoples should speak to histories of resistance and struggle, emancipation and success, sovereignty, and self-determination [41];What will the effects of the learning objective be? The outcome of the learning objective in a health context and whether it is towards cultural safety or oppressive segregation. 

Leonardo [70] additionally identifies questions pertaining to whiteness. Whiteness is not the same as White people; Whiteness is a racial discourse: the beliefs, practices, and assumptions that centre White people into positions of dominance and superiority [74]. While Whiteness theory is outside of the scope of this project, these questions are readily adaptable and sit well alongside CRT in both purpose and identifying power structures in raciology. They are:Is there avoidance of identifying with a racial experience or group, “othering” present within the learning objective? This question identifies whether “race” as a social construct is present within the objective;Who is placed as inferior and superior in the social construction of race within the learning objective? This is related to segregation based on race;Is inequity explained by reference to any number of alternative factors rather than being attributable to dispossession and colonisation by those who belong to Whiteness?

The questions will be presented in the following order as presented in Appendix A: Who wrote the learning objective?Is there avoidance of identifying with a racial experience or group, “othering” present within the learning objective?Who is placed as inferior and superior in the social construction of “race” within the learning objective?Who wins and who loses based on the priorities in the learning objective?Is inequity explained by reference to any number of alternative factors rather than being attributable to dispossession and colonisation by those who belong to Whiteness?What will the effects of the learning objective be?

## 4. Discussion

The discussion section of the paper will present the findings of the data analysis through a narrative method of counter-story telling as per the third tenet of ColonialCrit [42]. The power of qualitative data lies with the richness of the words [42]. Data will not be quantified but critically analysed, compared, and presented as such [42]. This research thus aims to uncover continuing forms of oppression, maintain political integrity in Indigenous research, and provide Indigenous voices within research, as per Rigney [44], through offering a counter story to the dominant narrative through my Indigenous paradigm. Through targeting “race” as a tool of oppressive segregation in curriculum, the ultimate goal of this research is to recommend that curriculum that is evaluated as leading to the reinforcement and teaching of racist ideologies to nursing students be modified to ensure it is not perpetuating racism among future health professionals, who have the social and professional responsibility of promoting culturally safe care. 

## 5. Ethics

This research, as a document analysis of publicly available data, does not require ethical approval since it does not involve humans, their data, or tissues/fluid [75]. An application has been made to the University of South Australia Human Ethics Research Committee, which has been considered and formally documented as exempt (application no: 203731). However, the project aligns with the six core values outlined within the *Ethical conduct in research with Aboriginal and Torres Strait Islander Peoples and communities: Guidelines for researchers and stakeholders* [76]: spirit and integrity, cultural continuity, equity, reciprocity, respect, and responsibility. The research is for us, as determined by the principles of Indigenous research by [44], rather than research on us or about us. The project actively works towards social equity as outlined in an array of articles in Universal Declaration on Bioethics and Human Rights [77] such as Article 11—Non-discrimination and non-stigmatization; Article 12—Respect for cultural diversity and pluralism; Article 14–19—Social responsibility and health; and Article 15—Sharing of benefits. The formal documentation of ethical exemption will enable me to contact universities who do not list their learning objectives publicly online. No universities will be identified throughout the study. This is to ensure anonymity of those universities that do not have their learning objectives publicly available. Data will be stored on a secure server on the University of South Australia network in a password protected and secure location. 

Whilst the research does not include Aboriginal or Torres Strait Islander participants, ethical considerations are still required including social justice and equity for Aboriginal and Torres Strait Islander peoples [78]. As a Ngarrindjeri woman working from an Indigenous research paradigm towards resistance, political integrity, and fore-fronting our voices, I am ethically accountable to my peoples [78]. This is the reason for my choice of an Indigenous research paradigm and the methodological decisions that were made based on the paradigm [78].

## 6. Conclusions

This study protocol highlights the critical need for anti-racist curriculum reform as a means to work towards care that can be received as culturally safe. Through the utilization of questions to assess educational policy and Whiteness, a tool is able to be formed that can assist nurse educators to critique their learning objectives and work towards anti-racist education within undergraduate nursing programs at Australian Universities. 

## Figures and Tables

**Figure 1 ijerph-19-07703-f001:**
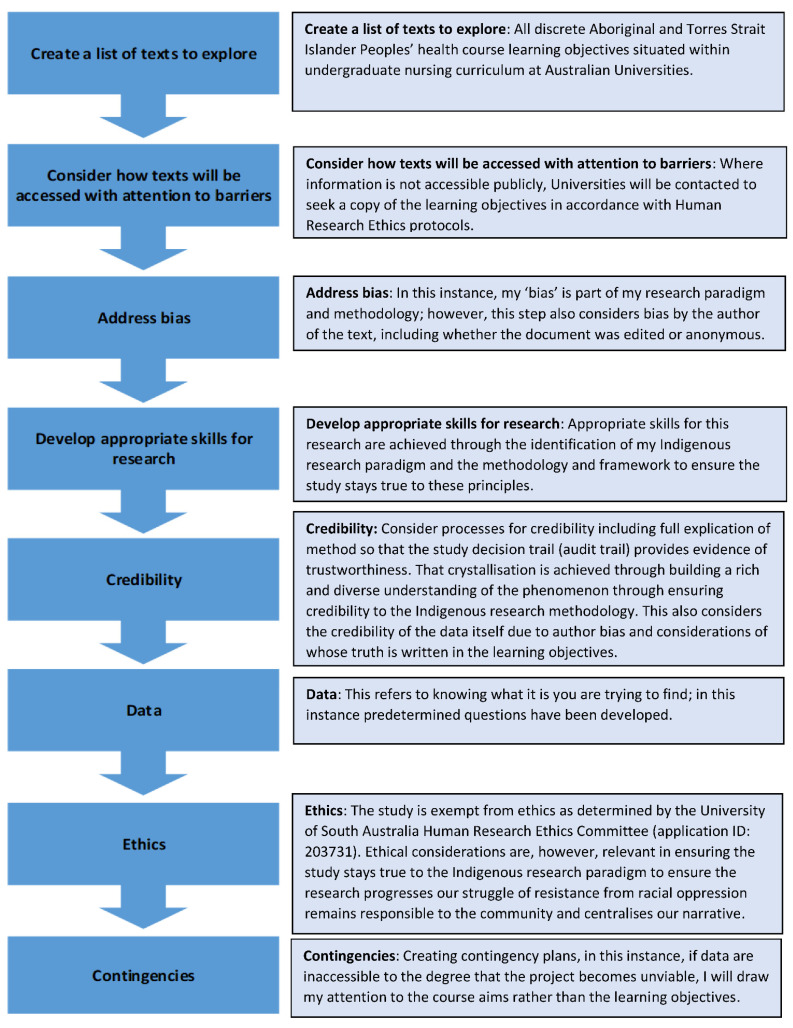
Eight-step planning process for textual analysis adapted from O’Leary [72].

## Data Availability

Not applicable.

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
