# Peer review of "Are Australian Universities Perpetuating the Teaching of Racism in Their Undergraduate Nurses in Discrete Aboriginal and Torres Strait Islander Courses? A Critical Race Document Analysis Protocol"

_ijerph, 2022, doi:10.3390/ijerph19137703_

Round 1

Reviewer 1 Report

Dear Editor,
I really appreciate the opportunity to review the manuscript ijerph-1662563 entitled:
"Are Australian Universities perpetuating the teaching of racism in their undergraduate nurses in discrete Aboriginal and Torres Strait Islander courses? A critical race document analysis protocol"

I commend the authors for describing this critical and timely issue. The paper is interesting and well-written; however, I would like to highlight some issues that merit revision:

First of all, I have noted some typos in the text, e.g. "utlisation", and so on; please, correct. 

Although the article is very well written, and the reading flows well, as the authors rightly point out in the limitations, a biased selection is present, prompting calls for reevaluation of the title.

"Are Australian Universities perpetuating the teaching of racism 2 in their undergraduate nurses in discrete Aboriginal and Torres 3 Strait Islander courses"

"Perpetuating" seems too strong as a tone and in these times of high social conflict, I recommend something less strong, for example instead of "perpetuating" I would suggest "allowing", "tolerate" weakly contrasting" and so on

Reviewer 2 Report

I read this submission with great interest and have some minor comments if the authors may want to consider.

As an international reader, I was wondering if there are other similar studies and work beyond US-Australia as in Europe and other continents, there are many indigenous populations exposed to severe discrimination and stigma in multiple contexts. The Roma minority across EU and beyond is a prime example of this, consistently subjected to intuitional racism and social marginalization.

Beyond regional traits, the covid-19 pandemic exacerbated these mechanisms and the authors may want to give more insights in the local context also in relation to the history of indigenous populations in Australia.

I would be curious to see an outline of the framework as applied and useful to other contexts and nations with indigenous groups as well as broader applications to racialized societies globally. What are the main implications globally?

Reviewer 3 Report

The Authors must see my remarks

Reviewer 4 Report

Thank you for your contribution to the methodological literature. Minimal revisions required.

See below revisions on grammar, sentence structure, and clarity. 
Line 15: 'W' removed but no 'w' added, sentence now reads as "here racism and...". if intentional, ignore this comment. 
Line 51: duplicate 'the'.
Line 52: sentence clarity, currently reads as "over all others and 21 thereby".
Line 119: Space needed between reference [28] and 'the'.
Line 128: Two spaces between 'personal' and 'and'.
Line 169: Two spaces between 'of' and 'colonization'.
Line 183: Unclear sentence revise, "...if race is used a tool to oppressively segregate..."
Line 236: hyphen needed for non-Indigenous
Line 245-8: Sentence requires clarity or reformatting. "Colonial Critical Race Theory methodology and framework Framing this research 246 within an Indigenous research paradigm has determined my choice in using Colonial Crit- 247 ical Race Theory (ColonialCrit)."
Line 255: sentence clarity, currently reads as "...1970's 17 and was...".
Line 299: Space needed between '[71].ColonialCrit'.
Line 342: sentence clarity, currently reads as "...2020- 12 2025 [26], to ensure...".
Line 359: Figure caption needs to be aligned with figure.

Review for other grammatical and clarity issues throughout.

Author Response

We would like to thank you and the reviewers for taking the time to review our manuscript, “Are Australian Universities perpetuating the teaching of racism in their undergraduate nurses in discrete Aboriginal and Torres Strait Islander courses? A critical race document analysis protocol”, and for the reviewers’ insightful feedback.  

We note this second round of reviews includes comments from Reviewer 3 and Reviewer 4 only.  We have already responded to all comments provided by Reviewer 3 in the first round of reviews (comments in round 1 and 2 from Reviewer 3 were identical). Therefore, we restrict our responses here to comments from Reviewer 4.  We thank the reviewer for highlighting these issues, and all suggested edits have been undertaken in the revised version of the manuscript. 

In addition, we wish to amend a response to Reviewer 2 from the first round of reviews.  This comment related to changing the term “perpetuating” to “allowing” in the manuscript title. We originally agreed with this change in our original response. However, on reflection we feel it is important to maintain the word ‘perpetuating’ for the title. We feel that the term ‘allowing’ lends towards passivity, whereas the term ‘perpetuating’ identifies that making no action to counter racism is an action in and of itself, that even unintentional racism is still racism.  In addition, the term is used as part of a question – it is important to critically evaluate this question with regard to health curriculum.  Therefore we will retain the original manuscript title, “Are Australian Universities perpetuating the teaching of racism in their undergraduate nurses in discrete Aboriginal and Torres Strait Islander courses? A critical race document analysis protocol”.

We believe that this revised version of the manuscript is improved on the original, given the incorporation of the considered comments and positive feedback from the reviewers.

Round 2

Reviewer 3 Report

The Authors must see my remarks

Author Response

(The authors gave the same response as above.)
